

# Integrated single-dose kinome profiling data is predictive of cancer cell line sensitivity to kinase inhibitors

Chinmaya U. Joisa[1], Kevin A. Chen[2], Matthew E. Berginski[3], Brian T. Golitz[4], Madison R. Jenner[3,5], Gabriela Herrera Loeza[5], Jen Jen Yeh[2,3,5] and Shawn M. Gomez[1,3]

[1] Joint Department of Biomedical Engineering, University of North Carolina at Chapel Hill and North Carolina State University, Chapel Hill, NC, United States of America
[2] Department of Surgery, University of North Carolina at Chapel Hill, Chapel Hill, NC, United States of America
[3] Department of Pharmacology, University of North Carolina at Chapel Hill, Chapel Hill, NC, United States of America
[4] Eshelman Institute for Innovation, University of North Carolina at Chapel Hill, Chapel Hill, NC, United States of America
[5] Lineberger Comprehensive Cancer Center, University of North Carolina at Chapel Hill, Chapel Hill, NC, United States of America

Corresponding author
Shawn M. Gomez, smgomez@unc.edu

## ABSTRACT

Protein kinase activity forms the backbone of cellular information transfer, acting both individually and as part of a broader network, the kinome. Their central role in signaling leads to kinome dysfunction being a common driver of disease, and in particular cancer, where numerous kinases have been identified as having a causal or modulating role in tumor development and progression. As a result, the development of therapies targeting kinases has rapidly grown, with over 70 kinase inhibitors approved for use in the clinic and over double this number currently in clinical trials. Understanding the relationship between kinase inhibitor treatment and their effects on downstream cellular phenotype is thus of clear importance for understanding treatment mechanisms and streamlining compound screening in therapy development. In this work, we combine two large-scale kinome profiling data sets and use them to link inhibitor-kinome interactions with cell line treatment responses (AUC/IC$_{50}$). We then built computational models on this data set that achieve a high degree of prediction accuracy (R$^2$ of 0.7 and RMSE of 0.9) and were able to identify a set of well-characterized and understudied kinases that significantly affect cell responses. We further validated these models experimentally by testing predicted effects in breast cancer cell lines and extended the model scope by performing additional validation in patient-derived pancreatic cancer cell lines. Overall, these results demonstrate that broad quantification of kinome inhibition state is highly predictive of downstream cellular phenotypes.

## INTRODUCTION

Computational drug screening has recently emerged as a powerful approach to integrate vast amounts of cancer cell line multi-omics data into predictive models, with the goal of predicting downstream phenotypic responses such as growth and viability. Advancing rapidly with recent developments in machine learning, these methods have the potential to predict outcomes for large drug libraries with minimal experimental cost, reducing the number of drug candidates fed into downstream validation efforts. Most current approaches to the prediction of drug response use baseline cell-line multi-omics data (*e.g.*, mutation status, gene expression, copy number variation, *etc.*) and quantitative structure-activity relationships (QSAR) to map drug structure characteristics onto their biological phenotypes. However, information describing drug-target interactions, especially at the protein level, remains underutilized due to the unique nature of associated data acquisition methods. For example, the recent DREAM challenge (*Costello et al., 2014*) hosted by the National Cancer Institute (NCI) for drug-response predictions saw the winning team utilize high throughput drug screening data along with baseline gene expression features (*Gönen & Margolin, 2014*) and achieved at most 80% accuracy in predicting cell line responses in a binary, yes/no fashion.

As a central component of cellular information transfer, protein kinases are enzymes that have also shown (*Gambacorti-Passerini et al., 2011*) promise as therapeutic targets, with initial success being found through the development of Imatinib (Gleevec). Drugs that inhibit kinases ("kinase inhibitors") are now one of the fastest growing clinical drug classes (74 FDA approved as of 2022), but around 1/3rd of all known kinases still have relatively unknown function and few chemical tools exist to interrogate and expand this knowledge. To explore the potential of the kinome as a therapeutic target, recent work has focused on profiling the full breadth of targets for kinase inhibitors, especially since many inhibitors have significant off target effects as a result of targeting the conserved ATP-binding pocket. Continued improvements in high-throughput assays such as Kinobead/MS (*Reinecke et al., 2019*), KINOMEscan (©DiscoverX), and KiNativ (*Patricelli et al., 2011*) now enable measurement of a given inhibitor's interactions across 250–500 kinases, providing a snapshot of its effect on the physiological kinome. We refer to this kinome-wide profiling data as the "kinome inhibition state" of a given inhibitor. This ability to generate drug-target interaction data on a large scale for a compound class is relatively unique, providing a novel means to leverage knowledge of off-target effects for drug response prediction.

The DepMap portal database (*Corsello et al., 2020*) contains thorough multi-omic characterization of ~1,000 cancer cell lines of all types, and corresponding cell viability measurements for about ~1,500 repurposed compounds, ~250 of which are kinase inhibitors. Using this data, we can connect kinase inhibitor phenotypes of cell viability to their "kinome inhibition states" and build models to predict the cellular responses to treatment with different kinase inhibitors. We have previously shown that these kinome states obtained solely through the kinobeads assay for clinical inhibitors are highly predictive of cancer cell viability, and also validated these predictions experimentally (*Berginski et al., 2022*). However, the kinobeads assay is unique and requires dedicated lab personnel
to run, restricting its use to relatively few labs. In contrast, the KINOMEscan assay is a popular and easily accessible commercial alternative that assays a panel of ∼500 native and mutant kinases recombinantly. Large amounts of KINOMEscan data have been deposited online by various groups (*Koleti et al., 2018*; *Wells et al., 2021*), including data for inhibitors developed against understudied kinases. These altogether account for four times as many inhibitors profiled (∼800) when compared to the data available from the kinobeads assay, representing a massive expansion of publicly available inhibitor state data. However, due to the uncharacterized nature of the inhibitors in the large KINOMEscan data set, only a small number of them (∼40) have been tested in the DepMap screening database, compared to ∼200 inhibitors from the kinobeads set.

In this work, we describe a framework to create an integrated kinome inhibition state data set by combining kinobeads and KINOMEscan data, and then leverage the entirety of this data into predictive models. This combined set contains single-dose inhibitor profiling data for a total of ∼800 kinases and kinase interacting proteins, spanning almost 1,000 kinase inhibitors that target a diverse section of the overall kinome space. When leveraged within a machine learning framework, and supplemented with baseline gene expression data, we are able to predict the sensitivity of ∼450 cancer cell lines in the DepMap screening dataset, with a reasonable $R^2$ of ∼0.7. Using this model, we were able to generate sensitivity predictions for 1.2 million inhibitor-cell line combinations, many of them targeted towards understudied kinases. We then experimentally validated these predictions in well characterized breast cancer cell lines seen by the model, as well as primary derived pancreatic cancer cell lines. We find reasonable agreement between predicted and observed outcomes in most compounds, seeing an expected drop in performance for understudied compounds and unique patient-derived cell lines. Together, these results show that there is a strong and predictive relationship between the state of the kinome (its "kinotype") and downstream cellular phenotypes, while further suggesting potential opportunities for leveraging computational models in inhibitor therapy design.

## METHODS

Portions of this text were previously published as part of a preprint (https://www.biorxiv.org/content/10.1101/2022.12.06.519165v1).

### Data sources

The primary data sources we used can be split into two categories: the integrated kinome profiling data set and the cancer cell line set:

The following were downloaded from the respective Supplementary Materials to create the integrated set of kinome profiling data:

1. Kinome profiling data from the kinobeads assay
   a. *Klaeger et al. (2017)*
2. Kinome profiling data from KINOMEscan assay
   a. LINCS: kinome profiling datasets for individual compounds downloaded programmatically from http://lincs.hms.harvard.edu/db/datasets/ (*Koleti et al., 2018*)

b. Kinome profiling data for the PKIS drug set was downloaded from the Supplementary Data of https://journals.plos.org/plosone/article?id=10.1371/journal.pone.0181585 (*Drewry et al., 2017*)

c. Kinome profiling data for the KCGS drug set was downloaded from the Supplementary Data of https://www.mdpi.com/1422-0067/22/2/566 (*Wells et al., 2021*) and from internal data provided by SGC-UNC.

The following were downloaded from the DepMap portal (https://depmap.org/portal/download/all/) to create the set of cancer cell line sensitivities and their gene expression characteristics:

1. DepMap secondary repurposing screen ("secondary-screen-dose–response-curve-parameters.csv")
2. CCLE gene expression set ("CCLE_expression.csv")

The high-throughput screening data gathered in PDAC patient-derived cell lines was gathered from *Lipner et al. (2020)* with methods as described in *Berginski et al. (2021a)*.

## Data preprocessing

The scripts implementing these descriptions are all available through github.

*Klaeger et al. Kinobead Kinase Inhibition Profiles*: As previously described (*Berginski et al., 2022*), we read the values from the Supplemental Data table into R and produced a filtered list of kinase and kinase interactor relative intensity values. We imputed missing values with the default "no interaction" value of 1, and truncated likely outlier values to the 99.99 percentile (3.43).

*KINOMEscan Inhibition Profiles*: We read in the three datasets mentioned above into R and concatenated them into a single combined data set. All the individual data sets contain identical protein lists because of the same assay type. Values are reported as "Percent Control", a ratio of protein pulled down in experimental condition (with inhibitor) *vs* control condition (without inhibitor). These were divided by 100 to convert the scale to 0–1 to match the Kinobeads relative intensity data.

*Creating the Combined Kinome Inhibition Profiling Set*: We took the kinobeads dataset and the KINOMEscan dataset and concatenated them into one large set containing inhibitor-kinase interaction states for ∼800 total kinases and kinase interactors. We left out assays that included recombinantly mutated kinases but left those with naturally occurring post-translational modifications. The vast majority (99.95%) of the inhibitor-kinase pairs represented was unique for either assay type, but for the 0.05% inhibitor-kinase pairs, we took the mean value of the measurements across the two assay types. Additionally, any missing values were imputed with the default "no interaction" value of 1. In the end we were left with kinome inhibition states for ∼1,000 kinase inhibitors.

*Dataset of Cancer Cell Line Sensitivity to Drugs from DepMap*: The DepMap repurposing dataset contains cell viability measurements across multiple doses, but since our dataset of kinome states is restricted to single-dose measurements, we extracted single summary statistics of cell line sensitivity to kinase inhibitors: Dose–response area under the curve (AUC) and half-maximal inhibitory concentration (IC$_{50}$). We extracted these by reading in the "secondary-screen-dose–response-curve-parameters" dataset into R, which contains

curve parameters for a log–logisitic curve fit to the cell viability dose response curve and filtered it down to cell line name, IC$_{50}$, AUC and other associated metadata.

*Matching of Kinase Inhibitors between Profiling Dataset and Cell-Line Sensitivity Dataset*: The compound names from each dataset were read into R, and the package Webchem (*Szöcs et al., 2020*) was used to retrieve PubChem compound IDs. The two sets of compound names were then matched based on these reference IDs. There were 252 matches between the two sets, forming a final set of ∼70,000 inhibitor-cell line combinations.

*Baseline Gene Expression*: As described before (*Berginski et al., 2022*) the RNAseq data provided in the "CCLE_expression.csv" file needed no modifications while preprocessing. Our only modification was to add identifiers to each gene label ("exp_"), to ensure that kinome inhibition data and expression data related to the same gene weren't accidentally combined.

*String*: The STRING database (*Szklarczyk et al., 2021*) was processed as described previously (*Berginski et al., 2022*) to annotate kinases and kinase interacting genes.

## Modeling techniques

To assess our models we used a random 10-fold cross validation strategy. The number of features was varied as specified by the feature selection scheme described in the results section. We compared the performance of three model types using this strategy: LASSO (Least Absolute Shrinkage and Selection Operator) regression using the glmnet engine (*Friedman, Hastie & Tibshirani, 2010*), random forest using the ranger engine (*Wright & Ziegler, 2017*) and gradient boosting using the XGBoost (eXtreme Gradient Boosting) engine (*Chen & Guestrin, 2016*). Model performance was assessed by the R$^2$ value between predicted and actual outcome within the cross-validation scheme. For each model type, we tuned sets of 30 hyperparameters to find the best possible performer as follows:

1. LASSO
   a. Penalty (1E−10–0.9)
2. Random Forest
   a. Trees (100–2000)
3. XGBoost
   a. Trees (100–1000)
   b. Tree Depth (4–30)

After final model selection, we fit the model on the entire dataset and then made predictions on inhibitor-cell line pairs not found in the original DepMap screening data.

## Compound testing

BT-474, HCC1806, SUM-159 and SKBR-3 cells were assayed as described previously (*Berginski et al., 2022*). Briefly, cells were grown in ATCC recommended media and seeded in triplicate at 4000, 2000, 4000 and 500 cells per well respectively. 24 h after seeding, cells were treated with inhibitors at 30 μM, 3 μM, 1 μM, 300 nM, 100 nM, 30 nM, 10 nM, and 3 nM, along with the appropriate DMSO controls. seeded at, in white flat-bottom 96-well plates (Corning). Seventy-two hours post-treatment, cells were lysed with CellTiter-Glo (Promega) and luminescence was read using the PHERAstar FS microplate reader (BMG

Labtech) and gain adjustments were conducted for each cell line. Notably, we used a 3-day treatment period instead of the 5-day treatment period used in the highly multiplexed PRISM assay, since we saw over-confluence in our 96-well plate assay in times longer than 3 days. However, it is likely that summary measures of trends in response like IC50 and AUC are minimally affected by the small difference in treatment times. Data were averaged over replicates normalized row-wise to the DMSO-only (0.1% on cells) control samples on each plate to calculate relative viability. Quality checks were performed to look at the data distribution and the presence of spatial bias on a plate. A quality control metric of <120% of DMSO was applied to all rows analyzed. The full file of cell viability curves is available in Table S1.

The functions "ComputeAUC" and "ComputeIC50" from the R package dr4pl (*Gadagkar & Call, 2015*) was used to fit a four-parameter log–logistic curve to the cell viability data, and extract AUC and $IC_{50}$ values from the 9-point cell viability curves.

## RESULTS

### Creating an integrated set of kinome profiling data across a wide chemical space

Kinase inhibitors have been profiled using a number of assays, but for this study we have focused on a specific subset of kinase inhibitors that have been assayed using the kinobead/MS-based method (*Klaeger et al., 2017*) or the KINOMEscan® (DiscoverX) method. These methods assess their specific kinase targets as well as the magnitude of inhibition of each kinase in response to different inhibitor concentrations (*Klaeger et al., 2017*). We combined kinome profiling datasets from *Klaeger et al. (2017)* (Kinobeads), LINCS (*Koleti et al., 2018*) (KINOMEscan), and UNC (*Drewry et al., 2017*) (KINOMEscan), filtering down to profiles measured only at 1uM. For the small amount of overlap between datasets, the mean inhibition value was taken across drug-kinase combinations. Given that both assays measure the engagement of inhibitors to kinases, most of the proteins that appear in assay results are either known kinases or closely associated proteins. Specifically, kinase inhibitor profiles include measurements on all wild-type and phosphorylated kinases (~500), along with a set of associated proteins (~300). As such, we will refer to this data as "kinase inhibition states", and the profile of each individual drug as its "kinome inhibition state" (Fig. 1A).

After integration, we were left with a final set of ~1,000 compounds with corresponding information on their inhibitor-induced kinome states, describing changes in ~800 kinases and kinase interactors. To summarize the relationship between inhibition states induced by all inhibitors, we performed a UMAP dimensionality reduction (*McInnes et al., 2018*) on the dataset, only using kinases profiled in both assays for visualization (Fig. 1B). The UMAP coordinates represent the aggregate effect of each inhibitor on the kinome, *i.e.*, it is a representation of the uniqueness of its kinome inhibition state. Inhibitors that have similar effects on the kinome will have similar coordinates, while disparate inhibitors will have coordinates that are far apart. Using this, we can examine the diversity of kinome space targeting in our dataset, based on the origin of kinome profiling data. Our analysis

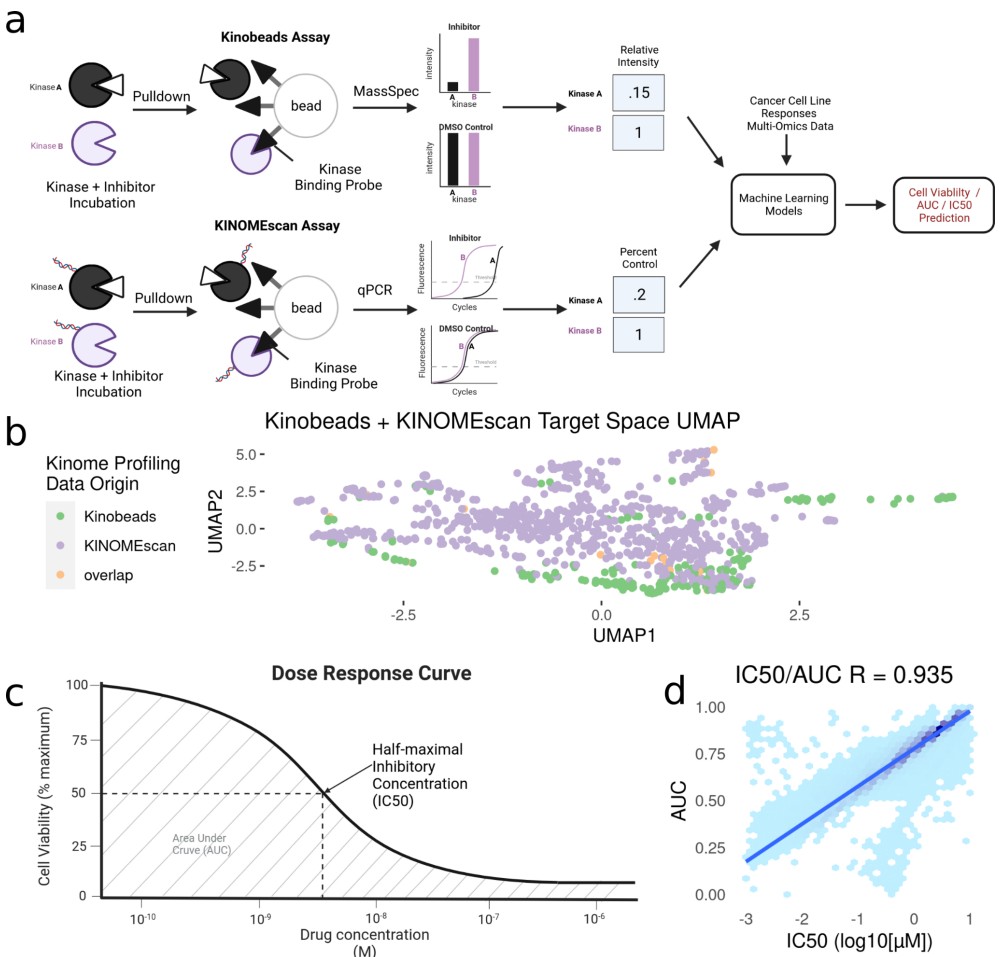

**Figure 1** **Modeling pipeline and target variable overview.** (A) Schematic of Kinobead/MS (upper) and KINOMEscan assay data integration into machine learning models predicting IC50 and AUC. (B) Visualization of UMAP dimensionality reduction on the combined kinome profiling data set, each point represents a single compound's position in the target space, and colors representing the origin of kinome profiling data. Target variables for modeling: (C) Extraction of IC50 and AUC from a drug's dose response curve in a given cell line. (D) Correlation and Scales of IC50 vs AUC values. Blue line indicates a linear model fit through the data.

shows that integrating KINOMEscan data for 800 inhibitors vastly increases the kinome space targeted (Fig. 1B), compared to kinobeads alone.

However, it is important to note that despite the similarity in the assay readouts, the KINOMEscan and kinobeads assays are very different molecularly, and care needs to be taken while integrating these assay types. To assess the degree of difference between the two assay types, we retained all kinases captured by both assays (∼250) and measured Pearson's correlation of inhibition states for the same compound between both assays. We found a reasonable degree of correlation (Pearson's R ∼0.5) across all data. In addition, we additionally found a much higher degree of agreement between data types (90%) when both assays were cut off at 80% inhibition (a "strong" hit in either assay) (Fig. S1). As a

result, keeping in mind the advantages of integrating both assay types, we moved forward with connecting these integrated kinome inhibition states to cell line responses.

## Connecting inhibited kinome states to cancer cell line sensitivities from the DepMap repurposing screen

To connect these kinase inhibitors and their induced inhibition states with their corresponding phenotypes in cancer cell lines, we make use of the DepMap repurposing screen, which uses the PRISM assay (*Yu et al., 2016*) to run highly multiplexed cell viability assays. This dataset contains cell viability measurements for over 1,500 drugs profiled in 450 cell lines. From within this data, we found ~200 drugs for which we also have corresponding profiling data as described above.

The DepMap repurposing dataset provides cell viability measurements across multiple drug doses. However, since our dataset of kinome states is restricted to single-dose measurements, we extracted two single summary statistics for describing cell line sensitivity to kinase inhibitors (Fig. 1C): Dose–response area under the curve (AUC) and half-maximal inhibitory concentration ($IC_{50}$). These properties are highly correlated with each other, having a Pearson's correlation coefficient ~0.9 (Fig. 1D). We extracted these properties from DepMap and matched them to our kinome states. The final integrated dataset has ~250 drugs tested across ~450 cell lines, representing ~70,000 inhibitor-cell line combinations representing nearly all cancer types.

## Examining bivariate association of features to cell line sensitivities provides a means for feature selection

The 450 cell lines tested in the DepMap dataset also have accompanying baseline RNAseq gene expression data, so we integrated the ~20,000 TPM values for each cell line into the kinome-state and cell line sensitivity dataset. This adds baseline cell line-specific gene expression information to our cell line agnostic inhibitor-induced kinome states.

To identify which features were most correlated with drug sensitivity, we examined bivariate associations of each of the ~21,000 features (kinome inhibition states, baseline gene expression, gene essentiality, protein expression and copy number variation) were compared individually to the outcome of cell line sensitivities (dose response AUC and $IC_{50}$), and their Pearson's R correlation was calculated. We additionally calculated Spearman's correlation coefficients, and ensured all correlated features were significant at the 0.5 level. The full table of feature correlations with significance values is attached in Table S2. We found that the most correlated feature is the drug-induced kinase inhibition state of TP53RK with a correlation coefficient R ~0.3, while the most correlated baseline gene expression value was OGFRL1 with a correlation coefficient R ~0.05. Overall, inhibitor-induced kinome states showed stronger correlation with cell line sensitivity metrics (Fig. 2A) despite there being 40× more baseline gene expression features than kinome states. However, it is important to note that this large difference in overall correlation may be partly due to the sample imbalance in abundance of kinome inhibition states (~200 per cell line) compared to baseline gene expression (one per cell line).

After exploring the relationship between features and cell line sensitivity, we sought to use machine learning models to integrate these features into predictive models of cell line

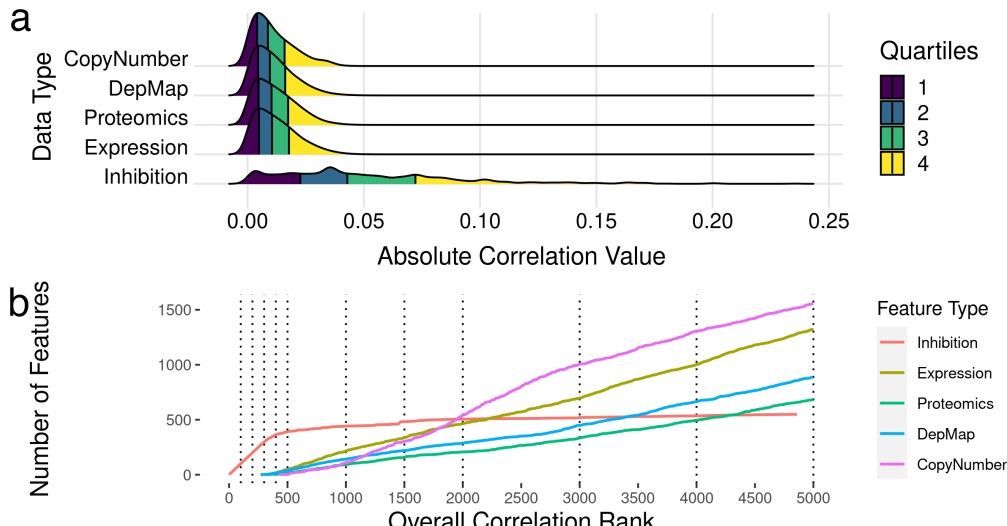

**Figure 2** **Feature selection by bivariate association with cancer cell line sensitivity.** (A) Ridgeline plot showing distributions of correlations with drug IC50s and AUC values across the data types included in analysis. (B) Plots showing what order classes of features are selected from the ranked set of inhibition states and baseline gene expression values. The dotted lines indicate the discrete increments of feature rank cutoffs at which model performance was tested. Kinase inhibition states were the most informative feature within the first ~300, after which gene expression features started to show predictive value.

sensitivity to kinase inhibitor treatment. To create a feature selection metric, all features were ranked by the absolute value of their Pearson's correlation to the cell sensitivity outcomes. Features were incorporated into models based on this ranking, *i.e.*, including only a top-ranked subset of best-correlated features. Seeking models with the best balance of predictive power and feature number, we tested model performance with different numbers of features, again selected starting from the most informative feature. We tested models with 100, 200, 300, 400, 500, 1,000, 2,000, 3,000, 4,000, and 5,000 features to see how adding more and more lower ranked features affected model performance. Figure 2B shows the ranking for all the feature sets, and the order they are added into the model.

## Machine learning models predict cancer cell line sensitivity from a combination of kinome inhibition states and baseline transcriptomics

To build machine learning models to predict cancer cell line AUC and $IC_{50}$ in response to treatment with kinase inhibitors, the highest ranked 100-5000 features were selected from the dataset linking drug-induced kinome states to cancer cell line responses (Fig. 2B). We compared three model types: LASSO regression, random forest and XGBoost. All models were trained with 10-fold random cross validation to minimize overfitting on the training data. Here, 10% of the dataset was randomly sampled and held out for testing, with the remaining 90% used for training. This was repeated ten times, and model performance was averaged across all the ten folds. This ensures comparable accuracy of the model predictions on new kinase inhibitors and cell lines. For each "batch" of feature numbers modelled from 100–5,000 we tuned sets of 30 hyperparameters for all model types (Fig. 3A). The $R^2$

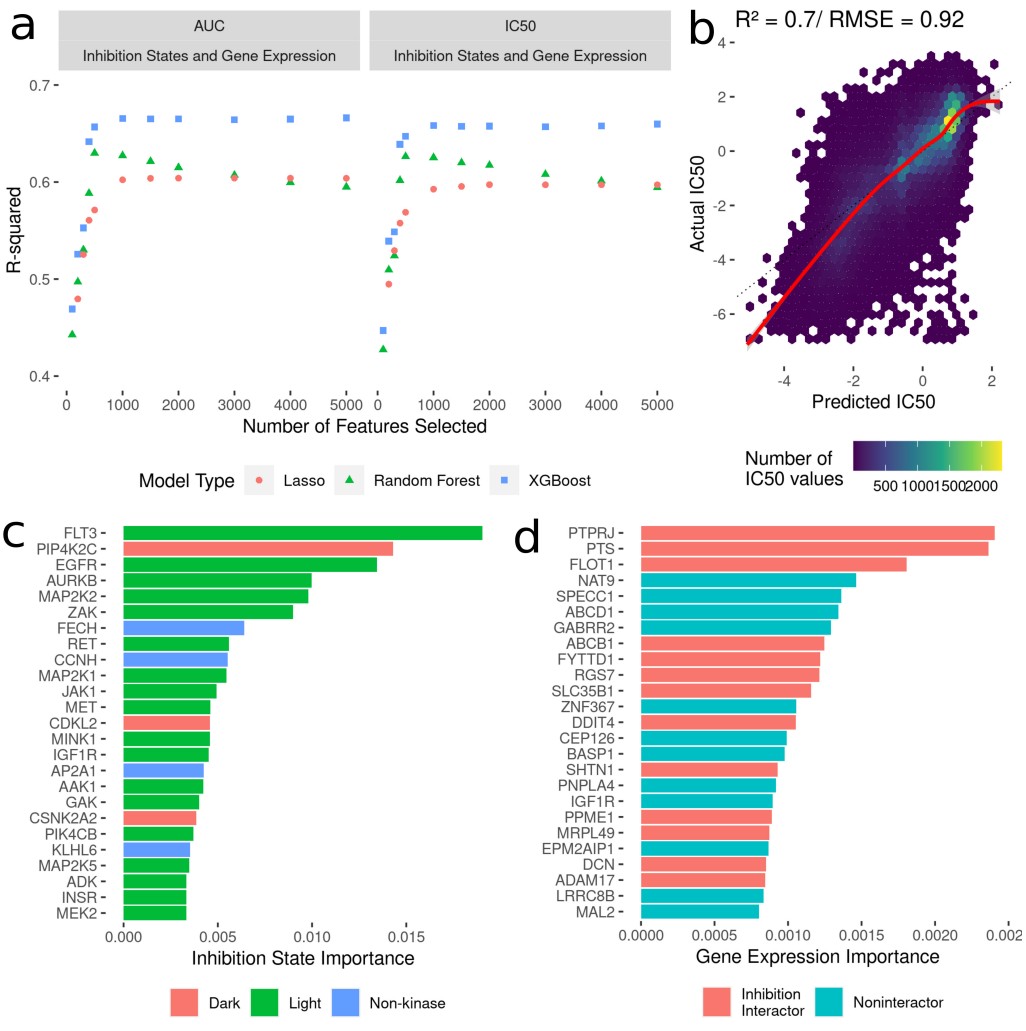

**Figure 3  Development of models to predict cancer cell line sensitivities to kinase inhibitors by integrating single-dose kinome profiling data.** (A) Model performance metrics ($R^2$) for LASSO (orange dot), Random Forest (green triangle) and XGBoost (blue square). (B) Scatterplot of predicted $IC_{50}$ values from the best-performing model vs actual $IC_{50}$ values. The red line indicated a smooth fit through the data points. (C) Horizontal bar plot showing model importance of individual kinase inhibition states by shapley values. (D) Horizontal bar plot showing model importance of individual baseline gene expression by shapley values.

value between predicted and actual value was utilized as the metric for model comparison. Overall, the 5000 feature XGBoost model performed the best with a cross-validation $R^2$ of ~0.7 (Fig. 3B).

Since tree-based machine learning models like XGBoost offer in-built explainability, it is possible to interrogate and explain which features were most important in predicting the outcome of cell line sensitivities. These importances generated *via* Shapley values (*Rozemberczki et al., 2022*) show kinase inhibition states to be overwhelmingly more important for predicting cell line responses when compared to baseline gene expression.

Kinases involved in cell cycle and proliferation are overrepresented in the top 25 features (Fig. 3C) (MAP2K, MEK2, CDKL5 *etc.*), and we further annotated each kinase with whether they are considered by current NIH guidelines as understudied ("Dark") or well-characterized ("Light") (*Berginski et al., 2021b*; *Oprea et al., 2018*). Interestingly four kinase interactor proteins are included as well, suggesting that interactions between inhibitors and non-kinases (off-target effects) have important consequences for cell viability. Baseline gene expression features show much lower model importances (Fig. 3D), but 40% of the top 25 genes have known interactions with kinases whose inhibition states are used in the model.

## Inclusion of various multi-omics data with kinome inhibition states and gene expression did not improve model predictive performance

In addition to the baseline gene expression data, all the cell lines in the DepMap database have three other profiling data types available: copy number variation, gene essentiality from CRISPR/KO, and baseline proteomics. To see if inclusion of these data into models would improve predictions, we integrated these with the modeling dataset of kinome inhibition states and gene expression and used identical modeling strategies described above to select correlated features, build, and evaluate LASSO, random forest, and XGBoost models (Fig. S2). We found that adding in the various multi-omic data types did not significantly outperform the models limited to kinase inhibition states combined with baseline gene expression ($R^2$ of ~0.69 for predicting $IC_{50}$).

## Experimental validation of model predictions were successful in characterized and novel cell lines

After fitting the model on 70,000 cell line-drug combinations, predictions were made on 1.2 million unseen (not previously seen by the model) drug-cell line combinations (Fig. 4A). Approximately 90% of the untested inhibitors were associated with KINOMEscan datasets. As an initial validation, we tested a subset of the predictions in well-characterized breast cancer cell lines, two unseen by the models and two previously seen (HER2 positive: SK-BR-3, BT-474 and two triple negative: SUM159, HCC1806). We analyzed the performance of the model on experimental data for unseen drug-cell line combinations, arriving at an R value ~0.6 for all but one (SKBR3) breast cancer cell line (Fig. 4B). Notably, all the drugs tested had kinome profiling data from the Kinobeads assay.

We then further validated the model by predicting inhibitor effects from collected RNAseq data in tumor (two samples) and stroma (one sample) derived cell lines from PDAC patients (*Lipner et al., 2020*; *Berginski et al., 2021a*). Importantly, these patient-derived cell lines were profiled for baseline gene expression in-house and represent a challenging and highly heterogeneous transcriptional landscape which the model has not seen before. Dose–response AUC predictions were made by the model for 58 drugs with kinome profiling data from the Kinobeads assay and 18 drugs with kinome profiling data from the KINOMEscan assay. The model predicted AUCs were compared to experimentally generated AUCs, revealing an average R ~0.5 for drugs with kinome profiling data from the Kinobeads assay tested in patient stroma-derived cell lines (Fig. 4C), and R ~0.4 for drugs with kinome profiling data from the KINOMEscan assay. On the other hand, in

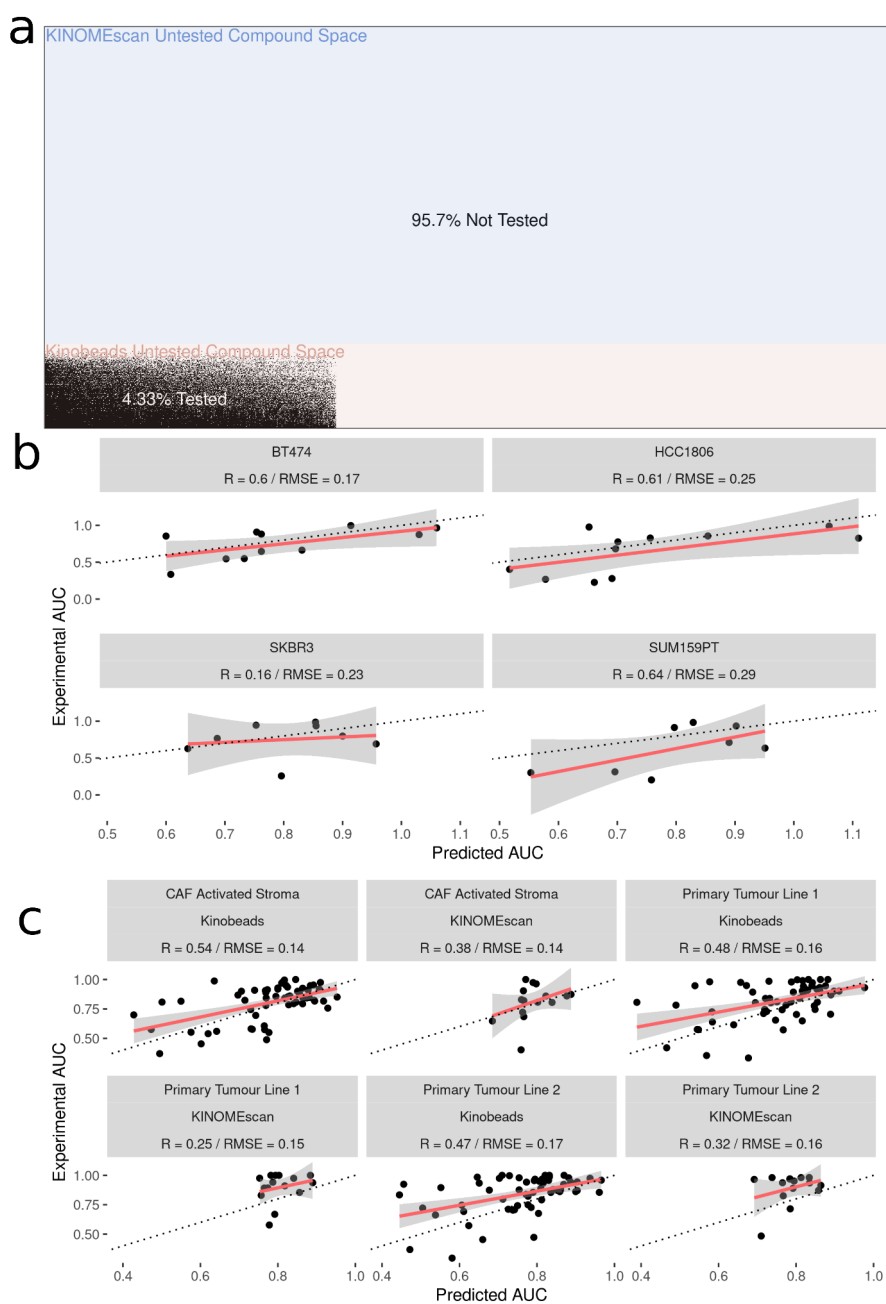

**Figure 4** **Experimental validation of model in breast cancer cell and patient-derived PDAC cell lines.**
(A) Visualization of the space of compound (*Y*-axis) and cell line (*X*-axis) combinations that have been tested (white) and not tested (black) with colors denoting the origin of the drug kinome profiling data as Kinobeads (Pink) and KINOMEscan (blue). (B) Scatter plot showing relationship between AUC's predicted by model and experimentally generated AUC's for drugs not yet tested by PRISM in breast cancer cell lines (C) Scatter plot showing relationship between AUC's predicted by model and experimentally generated AUC's for drugs from Kinobeads and KINOMEscan tested in primary-tumor and stroma PDAC cell lines.

patient tumor-derived cell lines, drugs from kinobeads had model accuracy R ~0.49, and drugs from KINOMEscan had R ~0.3 (Fig. 4C).

## DISCUSSION

Kinase inhibitors are one of the fastest growing classes of targeted cancer therapies, but only a small fraction of the druggable kinome has been explored to date (*Laufer & Bajorath, 2022*). We have previously used data describing kinobeads-derived kinome inhibition states to predict cell viability in cancer cell lines in response to clinical kinase inhibitors and shown high prediction accuracy. However, public databases of kinome inhibition states derived through the more accessible KINOMEscan assay cover many uncharacterized compounds and vastly widen the kinome space that can be targeted. In this work, we created a large integrated set of inhibitor-altered kinome states characterized by the kinobeads and KINOMEscan assays, representing a broad space of kinome targets and including a host of tool compounds targeting understudied kinases. We linked these kinome inhibition states to cancer cell line responses to kinase inhibitors (dose response AUC and $IC_{50}$) and then built machine learning models that integrate these kinome states with cell line baseline gene expression values to predict cell line responses to kinase inhibitors. Finally, we predicted cell line sensitivity to previously untested kinase inhibitors in breast cancer and patient-derived PDAC cell lines and validated them experimentally.

Prediction of therapy response for cancer cell lines has been demonstrated through various methods, mostly utilizing chemical structure information, baseline gene expression and gene mutation status. Drug-target interaction data is relatively under-utilized for phenotype prediction, but offers opportunities for biological hypothesis building, especially for compounds with uncharacterized mechanisms of action. Kinome profiling data provides an exciting opportunity to use a wide array of functionally relevant drug-target interactions, and is almost unique (except for GPCR inhibitors and HDAC inhibitors) in terms of ability to assay potential off target interactions. While we have previously shown that kinome profiling data generated from the kinobeads assay is informative for cell viability prediction, KINOMEscan data is more readily accessible, publicly available for many uncharacterized compounds, and easier to generate. In addition, integration of this data vastly broadens the kinome space capable of being targeted and increases the number of inhibitors that can be virtually screened as potential therapeutics.

It is important to note that this model linking kinome inhibition states to cell line response is generalizable to any human cancer sample, provided the sample has baseline transcriptomic data available. We have shown in this work and previously that models can reasonably extrapolate to kinase inhibitors that have not been tested before in well-characterized cell lines. Significantly, in this work we have extended the scope of the model by using novel RNAseq data from patient-derived PDAC cell lines, and testing kinase inhibitors against both tumor and stroma cell lines and achieving reasonable prediction accuracy.

Using tree-based models like gradient boosting lends us the ability to explain to some degree which features most influenced cell viability. Shapley importance values

generated from the best-performing model show that the inhibition states of kinases had overwhelmingly more predictive power compared to baseline gene expression values, with FLT3 as the most important feature. FLT3 mutations are observed in 30% of acute myeloid leukemia (AML) patients, and various FLT3 inhibitors are commonly prescribed for treatment (*Antar et al., 2020*). However, the study dataset contains a majority of cell lines from non-small cell lung cancer (NSCLC), and FLT3 inhibitors have recently shown promise in preclinical studies by abrogating DNA damage (*Ryu et al., 2019*). Although the baseline gene expression features had considerably lower predictive power in the model, it is important to note that they provided crucial cell line specific context to the model, essentially "tuning" the model for specific cellular contexts. Interestingly, 40% of all the top 50 gene expression features were annotated as known kinase interactors in the STRING database. An additional strength of modeling cancer response from kinase-drug interactions is the generation of new hypotheses for understudied kinases. For example, the understudied (*Essegian et al., 2020*) "Dark" kinases PIP4K2C and CSNK2A2 appear in the top 25 features of the best-performing model, suggesting possible functional roles in cancer cell viability. Interestingly, PIP4K2C expression has been associated with outcomes in Acute Myeloid Leukemia (AML) (*Lima et al., 2019*), while CSNK2A2 has been associated significantly with prognoses of 14 different cancer types (*Strum, Gyenis & Litchfield, 2022*).

There are still many limitations to the results reported in this study. The creation of a combined kinome profiling dataset involves gluing together results from different assay types. Although the assays produce the same output (ratio of kinase in treatment sample to kinase in control), there may be numerous methodological artifacts that add noise to the data. We have attempted to address this by analyzing model performance on each assay type individually, and we can see that responses to inhibitors with data originating from KINOMEscan are noisier and more difficult to predict than inhibitors with data from kinobeads. This discrepancy is potentially due to the imbalance in training data availability for the KINOMEscan inhibitors, with only 15 inhibitors having annotated cell line sensitivity data available. In the future, as more cell line sensitivity testing is performed for compounds in the KINOMEscan dataset, model performance for this assay may improve. Additionally, model performance also decreases when shifting from using gene expression data obtained from well-characterized cancer cell lines to using in-house gene expression of novel patient-derived cell lines. This is potentially due to the innate and significant heterogeneity that exists in such samples and because the models have been trained on baseline transcriptomics set of the given ~450 cell lines.

Additionally, there are important limitations to our initial analysis of correlations to cell line responses. While we observe higher correlations overall for kinome inhibition states, these are complicated by the sample imbalance in our abundance of drug-specific inhibition states (~200 per cell line) to gene expression values (one per cell line). This imbalance, characterized by the ratio of drug-specific kinome inhibition states to cell-line specific baseline gene expressions, is not only inherent to our study design but also reflective of the clinical scenario we aim to model. In a real-world setting, a patient is likely to have a single point of genomic profiling, most plausibly from a pre-treatment biopsy. Subsequent acquisition of multiple "cell-induced" genomic signatures in response

to a variety of treatments is not only challenging but also often impractical in a clinical setting. This underscores the importance and value of drug-specific descriptors, like the kinome inhibition states in our study, which can be obtained from high-throughput assays such as kinobeads and kinomescan. Consequently, while the imbalance does complicate interpretation, it also reinforces the relevance and applicability of our approach. We caution that the observed higher correlation for kinome inhibition states may be partly due to this sample ratio imbalance. Nonetheless, our primary measure of the relative importance of the data types is through the described multivariate machine learning analyses. We believe that these methods, which leverage the full complexity of the data, provide a more robust and comprehensive assessment of the information contained in each data type.

It should be possible to extend these models to incorporate multiple kinase inhibitors in combination. This is significant, given the frequency of resistance to targeted cancer monotherapies (*Lovly & Shaw, 2014*) and the potential to escape the resistance acquisition from dynamic kinome reprogramming through multi-inhibitor combinations (*Yesilkanal et al., 2021*). Thus, another area of future work is to combine the kinome inhibition states of multiple inhibitors to gain an understanding of their dual effect on the kinome, connecting them to biological phenotypes that arise in response to inhibitor combinations to eventually build models and predict effective kinome-targeting combination therapies.

While targeted therapies such as kinase inhibitors have had significant clinical impact, much work remains to better understand how modulation of the kinome leads to both desirable and undesirable phenotypic effects. The results presented here provide an example of one approach where knowledge of the inhibition state of the kinome can be linked to downstream phenotypes through predictive models, greatly enhancing our ability to predict the effects of existing inhibitor treatments as well as facilitating the design of novel targeted therapies in the future.

## ACKNOWLEDGEMENTS

We would like to thank Madison Jenner for providing data access for our PDAC validation. We would like to thank UNC Research Computing for access to the computational resources necessary for this work.

### Funding

This work was supported by grants through the National Institutes of Health (Grant #s CA274298, CA233811, CA238475, DK116204). The funders had no role in study design, data collection and analysis, decision to publish, or preparation of the manuscript.

### Grant Disclosures

The following grant information was disclosed by the authors:
National Institutes of Health: CA274298, CA233811, CA238475, DK116204.

## Competing Interests

Shawn M. Gomez is an Academic Editor for PeerJ.

## Author Contributions

- Chinmaya U. Joisa conceived and designed the experiments, performed the experiments, analyzed the data, prepared figures and/or tables, authored or reviewed drafts of the article, and approved the final draft.
- Kevin A. Chen conceived and designed the experiments, analyzed the data, authored or reviewed drafts of the article, and approved the final draft.
- Matthew E. Berginski conceived and designed the experiments, analyzed the data, authored or reviewed drafts of the article, and approved the final draft.
- Brian T. Golitz performed the experiments, authored or reviewed drafts of the article, and approved the final draft.
- Madison R. Jenner performed the experiments, authored or reviewed drafts of the article, and approved the final draft.
- Gabriela Herrera Loeza performed the experiments, authored or reviewed drafts of the article, and approved the final draft.
- Jen Jen Yeh analyzed the data, authored or reviewed drafts of the article, and approved the final draft.
- Shawn M. Gomez conceived and designed the experiments, analyzed the data, authored or reviewed drafts of the article, and approved the final draft.

## Data Deposition

The code is available at GitHub and Zenodo:

-https://github.com/gomezlab/kinomescan_viability_prediction.

-Chinmaya Joisa. (2023). Integrated single-dose kinome profiling data is predictive of cancer cell line sensitivity to kinase inhibitors. https://doi.org/10.5281/zenodo.8415100.

## Supplemental Information

Supplemental information for this article can be found online at http://dx.doi.org/10.7717/peerj.16342#supplemental-information.

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
