# Peer review of "Integrated single-dose kinome profiling data is predictive of cancer cell line sensitivity to kinase inhibitors"

_PeerJ, doi:10.7717/peerj.16342_

## Round 0.1 · original submission · Major Revisions

Dear Dr. Gomez,

Thank you for submitting your manuscript "Integrated single-dose kinome profiling data is predictive of cancer cell line sensitivity to kinase inhibitors" to PeerJ. We have now received reports from the reviewers, and, after careful consideration internally, we have decided to invite a major revision of the manuscript.

As you will see from the reports copied below, the reviewers raise important concerns. We find that these concerns limit the strength of the study, and therefore we ask you to address them with additional work. Without substantial revisions, we will be unlikely to send the paper back for review.

If you feel that you are able to comprehensively address the reviewers’ concerns, please provide a point-by-point response to these comments along with your revision. Please show all changes in the manuscript text file with track changes or color highlighting. If you are unable to address specific reviewer requests or find any points invalid, please explain why in the point-by-point response.

Thanks

Abhishek Tyagi, PhD
Academic Editor,
PeerJ

Reviewer 1 ·

Basic reporting

The paper proposed a computational model that predicts cell line treatment responses based on inhibitor-kinome interactions and cell expression data. The model is capable to identify kinases that affect cell responses, and is validated experimentally by testing predicted effects in breast cancer cell lines. Overall, the manuscript is innovative, along with reasonable experimental validation. However, the manuscript needs to keep on improving. The authors skipped many details which are needed to explain. The overall writing level also needs to be improved, such as the abstract needs to be further concise and optimized.

Experimental design

Regarding to the models of response prediction, the authors fail to effectively demonstrate what are the advantages of using inhibition state for improving model performance. In line 204, “the highest ranked 100-5000 features were selected from the dataset linking drug-induced kinome states to cancer cell line responses”, are the 100-5000 features including kinome states? What does the changes of model performance mean when adding features gradually into the models? In addition, as discussed in question 1, if the feature correlation with cell line sensitivity is not defined based on rigorous tests, the selected highest ranked 100-5000 features to cancer cell line responses may be unreliable. Overall, internally, the analysis should be presented more logically, along with additional evaluation metrics such as pearsonr, spearmanr and rmse; and externally, the comparison to other computational methods is necessary.

Validity of the findings

I have some concerns regarding bivariate association of features to cell line sensitivities. As kinase inhibition state is drug-induced, while CCLE gene expression is cell-induced, how to ensure an parallel comparison for their correlation against cell line sensitivity? It is unreasonable if concatenating the TPM values for the 450 cell lines to inhibitor-induced kinome states for ~1000 drugs, and then examined Pearson's R of each of the features against the dose response for all drug-cell pairs. The correlations between response and kinase inhibition states should be specific to a cell, and the correlations between response and cell expression values should be specific to a drug. However, by this way, it is hard to be compared equivalently. In addition, from figure 2a and 2b and their corresponding descriptions, it is hard to figure out how to calculate the R score and how to draw these red lines. Besides, the ticks of x-axis are incomplete and confusing.

Additional comments

In figure 1b, the UMAP shows that the distribution of target space of Kinobeads and KINOMEsacn are quite different. Is this because of the difference of their kinase panels? If so, it is better to show the overlap between their panels, as well as the sparsity of profiling data.

In “IC50”, “50” should be a subscript.

Reviewer 2 ·

Basic reporting

The manuscript is well-written overall. Below is a list of suggestions for some details to add/correct.
1. The citation and reference style should be adjusted to meet the PeerJ criteria.
2. Cell viability dose-response curves (measured by CellTiter-Glo) were not provided. The number of replicates performed was also not indicated.
3. Line 161: Check the placement of “(Fig. 1C)”. Reference to Fig. 1C should be earlier when describing the AUC and IC50?
4. Figure 2B should be described in a bit more detail.
5. “Dark” vs. “Light” Kinases should be described in more detail.
6. Line 216 and Figure 3 captions – capitalization of “S” in “Shapley”.
7. Figure 3C – check the definition of “non-kinase”. There appear to be kinases put under this classification, i.e., MEK2.

Experimental design

Having not extensively worked with XGBoost before, I am unable to provide an in-depth critique of the implementation of this machine learning algorithm. However, based on the information provided and a general understanding of statistics and ML principles, the selection of data and the construction of the model appear rational and justified. Additionally, the authors generated experimental validation data, including from patient-derived cancer cells – this contributes to demonstrating the broader applicability of their model. Nonetheless, below are a few additional comments.
1. The study could benefit from a more detailed explanation of the division of training, validation, and testing data.
2. In the validation experiments, the “Compound Testing” method section indicates a 72h (3-day) treatment time. According to the cited paper (Yu et al.) and description of the PRISM assay, the data appears to be generated by 5-day treatments. Although this would be unlikely to drastically affect the AUC, some explanation could help.

Validity of the findings

This is a very interesting study. The conclusions based on the presented data are overall reasonable, and the significance was not overstated. I also applaud the authors’ transparency in highlighting the limitations of their model in the discussions section. Below are some additional comments.
1. KINOMEscan vs. Kinobeads are different assays – they inherently will have different selectivity and sensitivity. The potential heterogeneity of the training data due to assay differences should be discussed more.
2. As the authors pointed out in Line 326, the majority of the cell lines in the dataset are lung cancer – could the model have a potential bias from possible overfitting to lung cancers? This is important and raises questions about the generalizability of this model in other cancer types and should be investigated and discussed more. I suggest taking an additional look at the correlation in non-lung cancers.
3. For the four cell lines tested in Figure 4B, both BT474 and HCC1806 were part of the cell lines included in the DepMap secondary repurposing screen used in the training data. If the data for these cell lines were used in training, they might not be representative of a true validation experiment, and the section should be worded more carefully.

---

## Round 0.2 · Minor Revisions

Dear Dr. Gomez,

Thank you for submitting your revised manuscript "Integrated single-dose kinome profiling data is predictive of cancer cell line sensitivity to kinase inhibitors" to PeerJ.

Based on reviewer comments, we decided to invite a minor revision of the manuscript.

Please kindly address the missing supplementary file and necessary correction as raised by the reviewer. Without substantial revisions, we will be unlikely to send the paper back for review.

Thanks
Abhishek Tyagi, PhD
Academic Editor,
PeerJ

Reviewer 2 ·

Basic reporting

The manuscript has improved since the first round of review. However, there are still some remaining comments on basic reporting that may need to be addressed.
- In the rebuttal letter, the authors mentioned, “We have added supplementary files (Table S2, S3, S4, S5) with the full dose response curves…”. However, these supplemental data could not be found. Please ensure the appropriate files for cell viability dose-response curves are correctly uploaded.
- Figure 3C’s categorization of dark, light, and non-kinases does not appear to be revised. Please recheck the categorization to ensure that they are accurate. MEK2 (MAP2K2) and PIK4CB were marked blue (non-kinases) in this version, which does not seem correct. It is also recommended to make reference to the specific “NIH guidelines” mentioned in lines 397-398.
- The Figure 2 caption in the PDF file was not updated, but the Figure caption in the text was updated.
- Reference style was not adjusted to PeerJ criteria.

Experimental design

The authors have provided adequate responses to my comments from the previous round of review.

Validity of the findings

The authors have provided adequate responses to my comments from the previous round of review.

---

## Round 0.3 · Minor Revisions

Dear Dr. Gomez,
Your article requires small Minor Revisions.

With kind regards,
Abhishek Tyagi

Reviewer 2 ·

Basic reporting

The authors have addressed the comments from the previous round of review.
Just one small point to check:
Line 386: “six kinase interactor proteins are included as well” – Figure 3C shows four non-kinases in the updated figure.

Experimental design

no comment

Validity of the findings

no comment

---

## Round 0.4 · accepted · Accept

Dear Dr. Gomez,
Thank you for your submission to PeerJ.
I am writing to inform you that your manuscript - Integrated single-dose kinome profiling data is predictive of cancer cell line sensitivity to kinase inhibitors - has been Accepted for publication.

Congratulations again, and thank you for your submission.

With kind regards,
Abhishek Tyagi
Academic Editor
PeerJ Life & Environment